# Concentrations of Co-Administered Meropenem and Vancomycin in Spinal Tissues Relevant for the Treatment of Pyogenic Spondylodiscitis—An Experimental Microdialysis Study

**DOI:** 10.3390/antibiotics12050907

**Published:** 2023-05-14

**Authors:** Josefine Slater, Maiken Stilling, Pelle Hanberg, Sofus Vittrup, Martin Bruun Knudsen, Sara Kousgaard Tøstesen, Josephine Olsen Kipp, Mats Bue

**Affiliations:** 1Department of Orthopaedic Surgery, Aarhus University Hospital, 8200 Aarhus, Denmark; maiken.stilling@clin.au.dk (M.S.); pellehanberg@clin.au.dk (P.H.); josephine.olsen@clin.au.dk (J.O.K.); matsbue@clin.au.dk (M.B.); 2Department of Clinical Medicine, Aarhus University, 8200 Aarhus, Denmark; 3Aarhus Denmark Microdialysis Research (ADMIRE), Orthopaedic Research Laboratory, Aarhus University Hospital, 8200 Aarhus, Denmark

**Keywords:** microdialysis, meropenem, vancomycin, spondylodiscitis, target site, spinal tissues, pharmacokinetics, pharmacodynamics, antibiotic stewardship

## Abstract

Co-administration of meropenem and vancomycin has been suggested as a systemic empirical antibiotic treatment of pyogenic spondylodiscitis. The aim of this study was, in an experimental porcine model, to evaluate the percentage of an 8-h dosing interval of co-administered meropenem and vancomycin concentrations above the relevant minimal inhibitory concentrations (MICs) (%T>MIC) in spinal tissues using microdialysis. Eight female pigs (Danish Landrace breed, weight 78–82 kg) received a single-dose bolus infusion of 1000 mg of meropenem and 1000 mg vancomycin simultaneously before microdialysis sampling. Microdialysis catheters were applied in the third cervical (C3) vertebral cancellous bone, the C3–C4 intervertebral disc, paravertebral muscle, and adjacent subcutaneous tissue. Plasma samples were obtained for reference. The main finding was that for both drugs, the %T>MICs were highly reliant on the applied MIC target, but were heterogeneous across all targeted tissues, ranging from 25–90% for meropenem, and 10–100% for vancomycin. For both MIC targets, the highest %T>MIC was demonstrated in plasma, and the lowest %T>MIC was demonstrated in the vertebral cancellous bone for meropenem, and in the intervertebral disc for vancomycin. When indicated, our findings may suggest a more aggressive dosing approach of both meropenem and vancomycin to increase the spinal tissue concentrations to treat the full spectrum of potentially encountered bacteria in a spondylodiscitis treatment setting.

## 1. Introduction

Pyogenic spondylodiscitis is a deep-seated infection of vertebral bodies and adjacent intervertebral discs [1]. The infection may also spread to involve other paraspinal tissues, and results in varying outcomes with relapse and mortality rates up to 18% and 23%, respectively [2,3,4]. Particularly alarming is the recent rise in the incidence of pyogenic spondylodiscitis in high-risk patient populations comprising elderly and other frail individuals with substantial comorbidities, intravenous drug abuse, and/or immunosuppressed patients [5]. Interestingly, the cases of culture-negative pyogenic spondylodiscitis are also increasing and are suggested to account for 7–42% of infected cases [6]. For culture-positive cases, the most common causative bacteria are *Staphylococcus aureus*, *Streptococcus species* (spp), *Enterococcus* spp., *Enterobacteriaceae* spp., and anaerobic bacteria [7].

The clinical diagnosis of spondylodiscitis is difficult, illustrated by an average diagnostic delay of up to two to four months, in addition to often unknown causative microbiology [8]. As a result, appropriate antibiotic treatment is often initiated late, and in many cases, relies solely on empirical treatment [3]. Current guidelines recommend initiating empirical treatment if the patients are hemodynamically unstable or have neurological deficits, and otherwise wait for microbiological diagnosis [1,9]. Still, the optimal choice for the empirical treatment of pyogenic spondylodiscitis remains unknown, including the dose and mode of administration, as demonstrated by alternating antibiotic regimens varying according to geography and local traditions [1,9,10]. Analyses of local resistance patterns should guide the choice of applied empirical antibiotics, but in many cases, this is lacking. For example, in Germany, an analysis of resistance patterns to applied empirical antibiotic regimens was performed [3]. Herein, 42% of pyogenic spondylodiscitis cases were inadequately covered, whereas a combination of meropenem and vancomycin displayed sufficient coverage for over 95% of the microbiological isolates. Similar findings were also demonstrated for other osteomyelitis cases [11]. As such, when empirical antibiotics are deemed necessary and local resistance patterns are lacking, a broad empirical antibiotic treatment encompassing both Gram-negative and -positive coverage appears necessary.

The selection of the optimal antibiotic treatment should, in addition to the drugs’ activity against the infecting bacteria, rely on the ability to reach sufficient concentrations, i.e., exceeding the minimal inhibitory concentrations (MIC) in the targeted tissues according to relevant pharmacokinetic/pharmacodynamic (PK/PD) indices. Thus, evaluating local antibiotic target tissue concentrations following common routes of administration and dosing are key when selecting drugs for the prevention and treatment of infections. Previously, the assessment of antibiotic concentrations in spinal tissue was challenging. In recent years, microdialysis has successfully been applied for the sampling of the free and active drug concentrations in spinal tissues of several antibiotic agents including vancomycin [12,13,14]. Still, meropenem spinal tissue concentrations are poorly described. In this experimental animal study, we set out to evaluate the tissue concentrations of systemically co-administered meropenem and vancomycin in the spinal tissues relevant for the treatment of spondylodiscitis, including the vertebral cancellous bone, intervertebral disc, paravertebral muscle, and adjacent subcutaneous tissue, during an 8 h interval. The primary endpoint was the percentage of an 8-h dosing interval with meropenem and vancomycin concentrations above relevant MICs (%T>MIC) for the bacteria most frequently encountered in pyogenic spondylodiscitis.

## 2. Results

All pigs (*n* = 8) completed the study. The data were obtained from all pigs except for one malfunctioning intervertebral disc catheter. The mean relative recoveries across compartments were 30–51% and 22–28% for meropenem and vancomycin, respectively. In the cases where the relative recovery could not reliably be determined, the mean relative recovery from the same tissue compartment and same drug was applied.

### 2.1. Meropenem

For both MIC targets, the T>MIC and %T>MIC of the 8-h dosing interval are presented in Table 1. The mean meropenem concentration-time profiles, and the relationship between the T>MIC and MIC targets, are graphically illustrated for all compartments in Figure 1. For the low MIC target (0.125 µg/mL), the %T>MIC was ≥54% for all the investigated compartments with the lowest value in the vertebral cancellous bone. For the high MIC target (2 µg/mL), the %T>MIC was ≤40% for all the investigated compartments, with the lowest value in the vertebral cancellous bone (25%).

The pharmacokinetic parameters are presented in Table 2. The meropenem tissue penetration (AUC_tissue_/AUC_plasma_) was the highest in the subcutaneous tissue 0.42, and lower but similar, ranging 0.21–0.27, in the intervertebral disc, vertebral cancellous bone, and paravertebral muscle.

### 2.2. Vancomycin

For both MIC targets, the T>MIC and %T>MIC of the 8-h dosing interval are presented in Table 1. The mean vancomycin concentration–time profiles and the relationship between the T>MIC and MIC targets are graphically illustrated for all compartments in Figure 2. For the low MIC target (1 µg/mL), the %T>MIC was ≥69% for all the investigated compartments with the lowest value in the intervertebral disc. For the high MIC target (4 µg/mL), the %T>MIC was ≤81% in all solid tissue compartments with the lowest values in the intervertebral disc (10%). 

The pharmacokinetic parameters are presented in Table 2. The tissue penetration (AUC_tissue_/AUC_plasma_) was incomplete for all tissues and ranged from 0.53–0.12, and was highest in the subcutaneous tissue and lowest in the intervertebral disc.

## 3. Discussion

We investigated the co-administered meropenem and vancomycin concentrations following a single-dose intravenous bolus infusion, in spinal tissues relevant for treatment of spondylodiscitis, in a porcine model. The main finding was that for both drugs, the %T>MICs were highly reliant on the applied MIC target, but heterogeneous across all targeted tissues, ranging from 25–90% for meropenem, and 10–100% for vancomycin. For both MIC targets, the highest %T>MIC was demonstrated in plasma, and the lowest %T>MIC was demonstrated in the vertebral cancellous bone for meropenem, and the intervertebral disc for vancomycin. Correspondingly, the tissue penetration of both drugs was incomplete in all the targeted tissue compartments.

Like other beta-lactams, meropenem’s bactericidal effect is commonly described as being related to the fraction of the dosing interval for which concentrations are higher than the MIC (T>MIC). Although an exact threshold for the treatment of spondylodiscitis is still controversial [15], a magnitude of at least 40%T>MIC is frequently cited [16]. Still, more aggressive %T>MIC targets (e.g., 100%T>MIC [17,18] and 100%T>4× MIC [19]) were shown to be necessary for microbiological and clinical cures in critically ill patients. Adhering to current clinical treatment guidelines [1], we administered meropenem as a bolus infusion of 1000 mg but found targeted tissue concentrations below the high MIC target of 2 µg/mL for most of the dosing interval (%T>MIC range: 25–36%). Similar findings were demonstrated for meropenem in cortical and cancellous tibial bone [20,21]. As such, these findings suggest that a meropenem bolus infusion of 1000 mg may not provide sufficient spinal tissue concentrations in an 8-h dosing interval to treat the full spectrum of potentially encountered bacteria in an empirical treatment setting. Equally important, it was shown for *Pseudomonas* spp. *in vitro* that the risk of acquiring meropenem-resistant bacteria colonization was higher when the %T>MIC was below 15% [22]. Thus, to achieve higher %T>MICs in spinal tissues, alternative dosing regimens appear necessary, e.g., the administration of a higher bolus dose up to 2000 mg three times daily, repeated administration, or extended/continuous infusion which is safely used for other infections and patient populations [23,24].

Despite vancomycin being extensively used for over 60 years, the most predictive PK/PD index is still widely discussed [25]. The target generally accepted is a plasma 24 h steady-state target ratio of AUC/MIC greater than 400 when the AUC is given in h×µg/mL [26]. Still, vancomycin treatment failure was demonstrated for up to 50% of patients suffering from methicillin-resistant *S. aureus* bacteraemia, illustrating a potential insufficiency of the current practice [27]. Further, it remains unclear if the current plasma target of AUC/MIC is applicable to the presented solid tissues [26]. Former animal studies attempted to correlate different vancomycin PK/PD indices with effect, i.e., T>MIC, AUC/MIC, and C_max_ /MIC, and found that the targeted tissue treatment effect against *Streptococcus* spp. and *S. aureus* was best described by T>MIC [28,29,30]. Therefore, in this study, we choose to present our results according to the (%) T>MIC. We found that vancomycin T>MIC for the high MIC target of 4 µg/mL was significantly lower in all targeted tissues than in plasma. These results support previous findings, reporting on MIC targets of 1–4 µg/mL in tibial cancellous and cortical bone [20]. Other than evaluating according to a certain PK/PD index, many factors, e.g., patient comorbidities, physiological differences, and infectious pathophysiology, can contribute to treatment insufficiency in the clinical setting. A previous experimental study indicated that vancomycin bone concentration decreased with the progression of the infection [31]. We administered 1000 mg in the present study. As recommended by most clinical guidelines, twice daily bolus infusions of 15–20 mg/kg up to 2000 mg appears to be well-tolerated by most patients [32]. Thus, to ensure the best possible vancomycin empirical treatment of spondylodiscitis, more aggressive dosing encompassing a loading dose or continuous infusion amongst other possible alterations may be needed to achieve higher spinal tissue concentrations. The use of a vancomycin loading dose is currently practiced clinically in many locations, but the treatment guidelines fail to provide a standardised recommendation for this. Altogether, although theoretical, this target discussion highlights that bacteria-antibiotic-host interactions are complicated, and a one-parameter-fits-all approach may be misleading, supporting the increasing trend towards personalized treatment approaches, especially in complicated cases.

This experimental porcine study has several limitations. Most importantly, our spinal tissue concentrations were obtained in healthy porcine tissues following a single dose for 8 h (480 min). For vancomycin administration, a 12-h dosing interval is generally recommended. Therefore, our results overestimate the %T>MIC if correlated to a standard dosing interval. Further, the present setup does not permit direct comparisons to established steady-state treatment targets. On the other hand, our dataset is strengthened by dense sampling, which has been demonstrated to improve the ability to predict target attainment [33]. For future investigations, performing population pharmacokinetic simulations based on spinal tissue concentrations would provide important knowledge regarding the effects of higher doses and other modes of administration. Moreover, we did not obtain measurements from all potentially relevant spinal tissues. Keeping the diverse histopathologic nature of spondylodiscitis in mind, meropenem and vancomycin tissue concentrations in, e.g., the facet joints or spinal canal could have provided important information. Additionally, as with all microdialysis studies, the inherent limitations of the technique must be mentioned, e.g., uncertainties associated with calibration and chemical assays. Last, although pigs have been shown to resemble humans in terms of anatomy and physiology, inborn differences are present, e.g., the blood supply to the intervertebral disc [34]. Consequently, the next intriguing step is to perform clinical targeted tissue measurements in steady-state following more aggressive modes of administration, such as an extended/continuous infusion for meropenem, and a loading dose for vancomycin, and provide concentrations for the evaluation of the range of PK/PD indices.

In conclusion, we demonstrated heterogeneous spinal tissue %T>MICs for both drugs and the applied MIC targets following a single-dose bolus infusion of co-administered meropenem and vancomycin, with the lowest %T>MIC in the vertebral cancellous bone and the intervertebral disc for both drugs. When indicated, our findings may suggest a more aggressive dosing approach of both meropenem and vancomycin, including a higher dose, loading dose, or extended/continuous infusion, to ensure sufficient spinal tissue concentrations to treat the full spectrum of potential bacteria in a spondylodiscitis treatment setting.

## 4. Materials and Methods

This experimental animal study was performed at the Department of Clinical Medicine, Aarhus University, Denmark, and approved by the Danish Animal Experiments Inspectorate (license No. 2017/15-0201-01184). The European Union legislation for the protection of animals used for scientific purposes (Directive 2010/63/EU) was followed in this study. The animals were treated according to the ARRIVE guidelines.

### 4.1. Study Overview

Eight female pigs (Danish Landrace breed, weight 78–82 kg) were included. In general anaesthesia, microdialysis catheters were placed in the third cervical (C3) vertebral cancellous bone, the C3-C4 intervertebral disc, paravertebral muscle, and adjacent subcutaneous tissue. Plasma samples were obtained as a reference. Abiding by the recommendations of replacement, reduction, and refinement (the 3 Rs) [35] when conducting animal studies, the data sampled in extrathecal spinal tissues in the present study are derived from animals where lower extremity data were reported previously [20,36]. A single-dose of 1000 mg of meropenem (Bradex, FrostPharma AB, Danderyd, Sverige), and 1000 mg of vancomycin (Bactocin, MIP Pharma GmbH, Blieskastel, Germany) were simultaneously administered intravenously as a bolus infusion prior to sampling. The dose and mode of administration used were in accordance with the clinical practice at our institution. Sampling was conducted for 8 h.

### 4.2. Target Definition: Minimal Inhibitory Concentrations

Given the diversity of possible causative bacteria in pyogenic spondylodiscitis, we chose to investigate a range of MIC targets for both meropenem and vancomycin. We considered vancomycin as the relevant drug of choice for coverage against Gram-positive organisms, while meropenem would provide Gram-negative and anaerobic coverage.

The epidemiological cut-off (ECOFF) for *Pseudomonas aeruginosa* and *Enterobacteria* spp., defined by the European Committee on Antimicrobial Susceptibility Testing (EUCAST), was used to evaluate the PK/PD index time above MIC (T>MIC) for meropenem, and ECOFF values for planktonic *Staphylococcus aureus, Streptococcus* spp., and *Enterococcus* spp. were used to evaluate the PK/PD index T>MIC for vancomycin [37]. For meropenem, these MICs were reported in the range of 0.125–2 µg/mL; thus, we chose 0.125 µg/mL and 2 µg/mL to represent a low and high target, respectively. For vancomycin, these MICs were reported in the range of 1–4 µg/mL; thus, we chose 1 µg/mL and 4 µg/mL as the low and high targets, respectively.

### 4.3. Study Procedures

The sampling of meropenem and vancomycin solid tissue concentrations was performed using microdialysis, which has been explained abundantly in many antibiotic bone pharmacokinetic studies [12,13] and review articles [38,39] (Figure 3). The microdialysis catheters used were CMA 70 (membrane length 10 and 20 mm, molecular mass cut of 20 kDa; M Dialysis AB, Stockholm, Sweden). Through an anterolateral approach, the C2, C3, and C4 vertebrae were exposed. Using a 2 mm drill, a 25 mm drill hole was made into the C3 vertebra in which a 20 mm microdialysis catheter was placed. Afterward, with aid from a splitable microdialysis introducer (M Dialysis AB, Stockholm, Sweden), a 10 mm microdialysis catheter was positioned in the C3-C4 intervertebral disc, and a 20 mm microdialysis catheter was positioned in both the exposed paravertebral muscle and adjacent subcutaneous tissue. Correct positioning of the microdialysis catheters in the vertebra and the intervertebral disc was verified by fluoroscopy. The catheters were connected to precision pumps (CMA 107; M Dialysis AB, Stockholm, Sweden) and perfused with 0.9% NaCl solution at a flow rate of 1 μL/min.

To keep the pigs under general anaesthesia throughout the study, a continuous infusion of propofol (550–600 mg/h) and fentanyl (0.6–0.7 mg/h) (Propofol/Fentanyl, B. Braun, Fredriksberg, Denmark) was provided. The arterial pH and body temperature were continuously monitored and kept within the ranges of 7.40–7.55 and 36.5–39.0 °C (normal porcine homeostatic range), respectively. Glucose was administered when necessary and 0.9% NaCl was given continuously (150 mL/h), maintaining normohydration. The pigs were euthanised with intravenous pentobarbital at the end of the study.

### 4.4. Tissue and Plasma Sampling

After catheter placement, microdialysis sampling was initiated with a tissue-equilibration period of 20 min prior to the antibiotic intravenous administration. Following microdialysis sampling, meropenem (1000 mg) and vancomycin (1000 mg) were administered intravenously over 10 min and 100 min, respectively. Microdialysates were collected at 30 min intervals for the first 4 h, and at 60 min intervals for the last 4 h, resulting in a total of 12 samples during an 8-h dosing interval. At the midpoint of each sampling interval, blood samples were collected from a central venous catheter for reference. The blood samples were stored at 5 °C for a maximum of 6 h before being centrifuged (room temperature, 3000 g, 10 min), and plasma aliquots were made. At the end of sampling and after 30 min of catheter-equilibration, retrodialysis by drug [40] was performed during a 40 min recovery sample using a 0.9% NaCl solution containing 100 µg/mL meropenem and 300 µg/mL vancomycin. The recovery sample was later used for individual catheter calibration and calculation of absolute tissue concentrations [40]. All collected samples were stored at –80 °C until the chemical analyses were performed; microdialysate analyses were performed at the Department of Forensic Medicine, Aarhus University, Denmark, and plasma aliquot analyses were performed at the Department of Clinical Biochemistry, Aarhus University Hospital, Aarhus, Denmark.

### 4.5. Chemical Analysis

The free drug concentration of meropenem in plasma was quantified using ultra-high performance liquid chromatography (UHPLC) [24]. The free drug concentration of vancomycin in plasma was quantified using a clinical standard homogenous enzyme immunoassay technique (Chemistry XPT, Advia Chemistry, Siemens Healthineers, Erlangen Germany) [21]. For the plasma samples, the lower limits of quantification (LLOQ) of both drugs were found to be 0.5 µg/mL. For meropenem, the accuracy of quantification was found to be between −4.3% and 4.8%, within a linearity range between 0.5 µg/mL and 105 µg/mL. For vancomycin, the intra-run imprecision (percentage coefficients of variation (%CV)) was 3.0% at 2.0 µg/mL. The free drug concentrations of meropenem and vancomycin in the microdialysates were quantified using tandem mass spectrometry (MS/MS) and UHPLC [17]. The method displayed satisfactory levels of precision (CV < 15%) in the quantification range between 0.1 µg/mL and 20 µg/mL.

### 4.6. Data Analysis and Statistics

A non-compartmental analysis was performed separately for each animal, each compartment, and for both meropenem and vancomycin. By linear interpolation for each compartment, T>MIC for MIC 0.125 and 2 μg/mL was estimated for meropenem, and T>MIC 1 and 4 μg/mL for vancomycin. The area under the concentration–time curves from zero to the last measured value (AUC_0-8h_) was calculated using the trapezoidal rule. The C_max_ was calculated as the maximum concentration of all the recorded concentrations, and the T_max_ was calculated as the time to achieve C_max_. As a measure of tissue penetration, the ratio of tissue AUC to plasma AUC (AUC_tissue_/AUC_plasma_) was calculated. A general comparison of the T>MIC and the pharmacokinetic parameters was conducted using repeated measurements of the analysis of variance (ANOVA), followed by pairwise comparisons made by linear regression. Because of a small sample size, it was necessary to conduct a correction for the degrees of freedom by the Kenward–Roger approximation method. Using the visual diagnosis of residuals, fitted values and estimates of random-effects model assumptions were tested. A *p*-value of less than 0.05 was considered significant. The calculation of pharmacokinetic parameters and statistical analyses were performed in Stata (v. 15.1, StataCorp LLC, College Station, TX, USA), and the calculation of T>MIC was performed in Microsoft Excel (version 16.66.1, Microsoft, Washington, DC, USA).

## Figures and Tables

**Figure 1 antibiotics-12-00907-f001:**
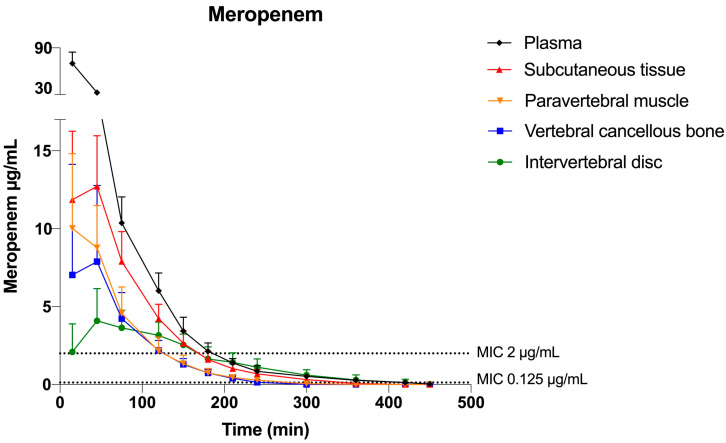
Meropenem mean concentration—time profiles for plasma, subcutaneous tissue, paravertebral muscle, vertebral cancellous bone, and intervertebral disc. Error bars represent the 95% confidence interval. The horizontal dotted lines indicate the high and low MIC targets of 0.125 and 2 μg/mL, respectively.

**Figure 2 antibiotics-12-00907-f002:**
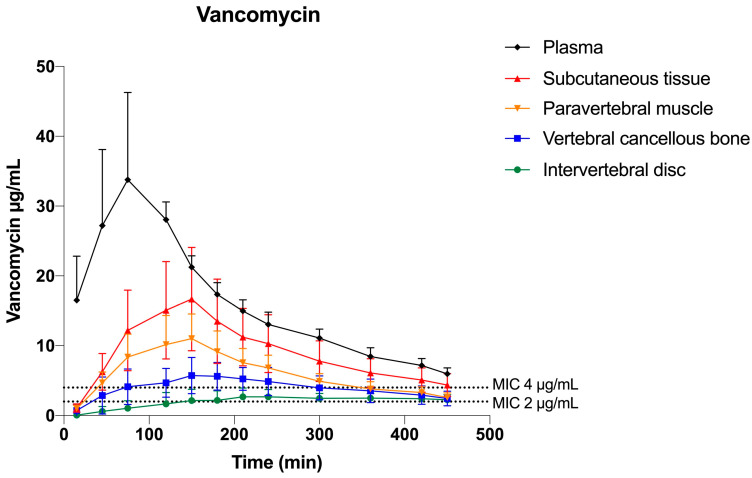
Vancomycin mean concentration–time profiles for plasma, subcutaneous tissue, paravertebral muscle, vertebral cancellous bone, and intervertebral disc. Error bars represent the 95% confidence interval. The horizontal dotted lines indicate the high and low MIC targets of 2 and 4 μg/mL, respectively.

**Figure 3 antibiotics-12-00907-f003:**
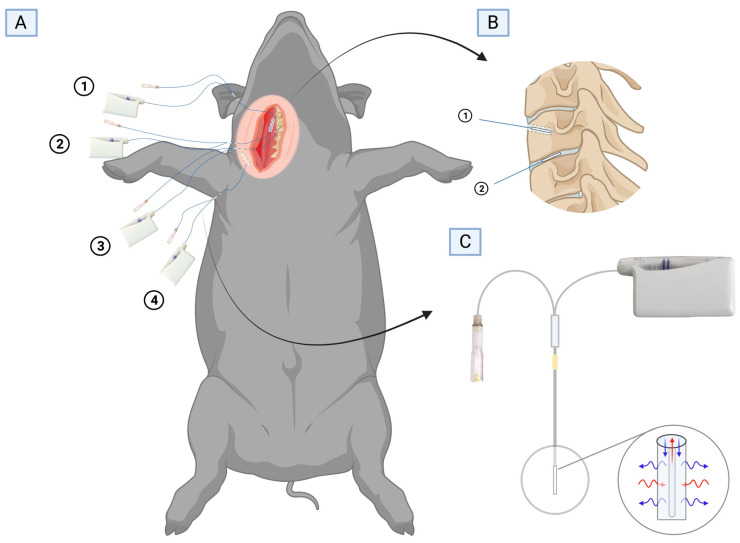
(**A**) A pig in supine position showing the area of incision and placement of microdialysis catheters in (1) the C3 vertebral body, (2) the C3–C4 intervertebral disc, (3) the paravertebral muscle, and (4) the subcutaneous tissue. (**B**) Magnification of the microdialysis catheter placement in (1) the C3 vertebral body and (2) the intervertebral disc. (**C**) Overview of the microdialysis system with a semipermeable membrane and inflow (perfusate) and outflow (dialysate). Created with BioRender.com.

**Table 1 antibiotics-12-00907-t001:** Time above MIC values for meropenem and vancomycin according to a low and high MIC target in separate tissue compartments, including plasma, subcutaneous tissue, paravertebral muscle, vertebral cancellous bone, and intervertebral disc.

Tissue Compartment	Mean T>MIC (min)	Mean T>MIC (min)	Mean %T>MIC	Mean %T>MIC
*Meropenem* Plasma	*0.125 μg/mL (low)* 406 (378; 435) ^a^	*2 μg/mL( high)* 181 (165; 198) ^c^	*0.125 μg/mL (low)* 90 (84; 97)	*2 μg/mL (high)* 40 (37; 44)
Subcutaneous tissue	334 (305; 363)	163 (147; 180)	74 (68; 81)	36 (33; 40)
Paravertebral muscle Vertebral cancellous bone Intervertebral disc	285 (256; 314) 244 (215; 272) ^b^ 366 (335; 396) ^e^	118 (101; 134) 114 (98; 131) ^d^ 158 (141; 175) ^e^	63 (57; 70) 54 (48; 61) 81 (74; 88)	26 (22; 30) 25 (22; 29) 35 (31; 39)
*Vancomycin* Plasma Subcutaneous tissue Paravertebral muscle Vertebral cancellous bone Intervertebral disc	*1 μg/mL (low)* 449 (406; 492) 435 (393; 478) 433 (391; 476) 388 (346; 431) ^d^ 311 (265; 356) ^e^	*4 μg/mL (high)* 446 (386; 505) ^a^ 365 (305; 424) 306 (306; 366) 214 (154; 273) ^b^ 45 (−19; 108) ^e^	*1 μg/mL (low)* 100 (90; 109) 97 (87; 106) 96 (87; 106) 86 (77; 96) 69 (59; 79)	*4 μg/mL (high)* 99 (86; 112) 81 (68; 94) 68 (68; 81) 47 (34; 61) 10 (−4; 24)

MIC, minimal inhibitory concentration; T>MIC, time with concentration above MIC; %T>MIC, percentage of dosing interval of 450 min with concentration above MIC. ^a^
*p* < 0.05 for plasma compared with all solid tissue compartments. ^b^
*p* < 0.05 for vertebral cancellous bone compared with subcutaneous tissue, paravertebral muscle, and intervertebral disc. ^c^
*p* < 0.05 for plasma compared with paravertebral muscle, vertebral cancellous bone, and intervertebral disc. ^d^
*p* < 0.05 for vertebral cancellous bone compared with intervertebral disc. ^e^
*p* < 0.05 for plasma compared with intervertebral disc.

**Table 2 antibiotics-12-00907-t002:** Key pharmacokinetic parameters for meropenem and vancomycin in plasma, subcutaneous tissue, paravertebral muscle, vertebral cancellous bone, and intervertebral disc.

Parameter	Meropenem	Vancomycin
*AUC_0-8h_ (min μg/mL)* Plasma	3044 (2815; 3273) ^a^	6789 (539; 2858) ^a^
Subcutaneous tissue	1267 (1038; 1496)	3767 (2608; 4927)
Paravertebral muscle Vertebral cancellous bone Intervertebral disc	808 (579; 1036) 705 (476; 934) ^b^ 655 (412; 899)	2538 (1379; 3698) 1698 (539; 2858) ^b^ 877 (−325; 2079)
*C_max_ (μg/mL)* Plasma Subcutaneous tissue	67 (59; 74) ^a^ 14 (6; 21)	35 (29; 41) ^a^ 17 (11; 23)
Paravertebral muscle Vertebral cancellous bone Intervertebral disc	10 (3; 18) 9 (2; 16) 5 (−3; 12)	11 (5; 17) 6 (0.4; 12) ^b^ 3 (−3; 10)
*T_max_ (min)* Plasma	15 (3; 27) ^c^	92 (45; 138) ^e^
Subcutaneous tissue Paravertebral muscle Vertebral cancellous bone	34 (22; 46) 30 (18; 42) 37 (25; 50) ^d^	135 (88; 181) 126 (79; 172) 174 (128; 221) ^d^
Intervertebral disc	34 (22; 46)	275 (225; 325)
*AUC_tissue_/AUC_plasma_* Subcutaneous tissue Paravertebral muscle	0.42 (0.35; 0.49) 0.27 (0.20; 0.34)	0.53 (0.41; 0.64) 0.36 (0.24; 0.47)
Vertebral cancellous bone	0.24 (0.16; 0.31) ^b^	0.24 (0.12; 0.35)^b^
Intervertebral disc	0.21 (0.14; 0.29)	0.12 (0.41; 0.64)

AUC_0-8h_, area under the concentration–time curve from 0 to 8 h; C_max_, peak drug concentration; T_max_, time to Cmax; AUC_tissue_/AUC_plasma_, area under the concentration–time curve ratio of tissue-free/plasma-free. AUC_0-8h_, C_max_, and T_max_ are given as mean (95%CI). ^a^
*p* < 0.05 for plasma compared with all solid tissue compartments. ^b^
*p* < 0.05 for the vertebral cancellous bone compared with subcutaneous tissue. ^c^
*p* < 0.05 for plasma compared with subcutaneous tissue, vertebral cancellous bone, and intervertebral disc. ^d^
*p* < 0.05 for vertebral cancellous bone compared with intervertebral disc. ^e^
*p* < 0.05 for plasma compared with vertebral cancellous bone and intervertebral disc.

## Data Availability

The data are available from the corresponding authors upon reasonable request.

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
