# Peer review of "Concentrations of Co-Administered Meropenem and Vancomycin in Spinal Tissues Relevant for the Treatment of Pyogenic Spondylodiscitis—An Experimental Microdialysis Study"

_antibiotics, 2023, doi:10.3390/antibiotics12050907_

Round 1

Reviewer 1 Report

Dear Editor,

I have reviewed the manuscript entitled “Concentrations of Co-administered Meropenem and Vancomycin in Spinal tissues relevant for the Treatment of Pyogenic 3 Spondylodiscitis – An experimental Microdialysis study” by Slater et al. The authors in this study evaluated the percentage of an 8-h dosing interval of coadministration one-gram meropenem and one-gram vancomycin above T<MIC in eight female pigs. The samples were taken by microdialysis from vertebral cancellous bone, intervertebral disc, paravertebral muscle, and adjacent subcutaneous tissue. Plasma samples were obtained for reference. The authors found the concentration of these antibiotics in vertebral samples were low and they suggested a more aggressive dosing approach of both meropenem and vancomycin to increase spinal tissue concentrations. The study is well designed and written.

Comments

Why do not the authors test their suggestion (increase the dose) to proof their conclusion?

Are these drugs safe when increase the dose above one gram? For example, Meropenem dose is 1 g IV q8hr; not to exceed 2 g IV q8hr

Minor comments:

Line 17: Abstract: Please change “8 female pigs” into “Eight female pigs”.

I have reviewed the manuscript entitled “Concentrations of Co-administered Meropenem and Vancomycin in Spinal tissues relevant for the Treatment of Pyogenic 3 Spondylodiscitis – An experimental Microdialysis study” by Slater et al. The authors in this study evaluated the percentage of an 8-h dosing interval of coadministration one-gram meropenem and one-gram vancomycin above T<MIC in eight female pigs. The samples were taken by microdialysis from vertebral cancellous bone, intervertebral disc, paravertebral muscle, and adjacent subcutaneous tissue. Plasma samples were obtained for reference. The authors found the concentration of these antibiotics in vertebral samples were low and they suggested a more aggressive dosing approach of both meropenem and vancomycin to increase spinal tissue concentrations. The study is well designed and written.

Comments

Why do not the authors test their suggestion (increase the dose) to proof their conclusion?

Are these drugs safe when increase the dose above one gram? For example, Meropenem dose is 1 g IV q8hr; not to exceed 2 g IV q8hr

Minor comments:

Line 17: Abstract: Please change “8 female pigs” into “Eight female pigs”.

Reviewer 2 Report

In a porcine model, the authors reported tissue and spinal concentrations of meropenem and vancomycin after a single infusion. The data showed acceptable concentrations for meropenem but clearly inadequate concentrations for vancomycin.

The model is well explained and the dosing and reporting of the data is very clear. 

Unfortunately, as mentioned by the authors, the model is performed in non-inflammatory tissue after a single dose in pigs and could not be considered representative of the concentration obtained in infected tissue in humans at appropriate doses and at steady state.

The authors claimed that the dose and route of administration were those used in spinal infections. However, most of the clinicians used the highest doses. Firstly, most of the time 30 mg/kg loading dose of vancomycin is administered in a 2 hour infusion, secondly, the use of continuous infusion of vancomycin after the loading dose is often preferred.  

Ultimately, I am not sure that the data presented here will help in the management of patients. Concentration obtained particularely with vanco were low but not representative of the usual way of treating patients and were performed in healthy, non-inflammatory pigs.

Reviewer 3 Report

The paper`s rationale is clear. However, the methodology needs a little further explanation. For instance, was there any sham surgery in this work? Why didn’t the authors try different concentrations of the tested drugs instead of a single concentration?

Minor comments:

Line 14-15: in an experimental model > in an experimental pig model

Line 17: 8 female pigs > Eight….  (Danish landrace breed, weight 78–82kg > weight = 78–182kg

Line 19-20: Microdialysis catheters were applied in vertebral cancellous bone: please name the vertebra here (third cervical or C3).

Line 52: In example > For example?

Line 74: percentages of an 8-dosing intervals > 8h?

Line 85, 98: T>MIC and %T>MIC: list in full. Don’t start paragraphs with abbreviations!

Line 230: C3 vertebral cancellous bone > third cervical (C3) vertebral cancellous bone

Line 329: C3 vertebral body > third cervical (C3) vertebral body

The authors should revise the use of the sub-headers 2.3 and 4.7.

The English language complies with academic standards and is understandable.

Round 2

Reviewer 2 Report

nothing to add.